

# Intermittent horizontal mattress suture in proximal anastomosis for acute type A aortic dissection: a retrospective study

Jiajun Li[1,*], Yongzhi Zhou[1,*], Yucong Zhang[2], Xuegui Chen[1], Jing Wang[1], Xiang Wei[1] and Min Hu[1]

[1] Division of Cardiothoracic and Vascular Surgery, Tongji Hospital, Tongji Medical College, Huazhong University of Science and Technology, Wuhan, Hubei, China

[2] Institute of Gerontology, Department of Geriatrics, Tongji Hospital, Tongji Medical College, Huazhong University of Science and Technology, Wuhan, Hubei, China

[*] These authors contributed equally to this work.

Corresponding author
Min Hu, Huminchn@tjh.tjmu.edu.cn

## ABSTRACT

**Objective**. To compare intermittent horizontal mattress suture (IHMS) technique and conventional sandwich technique for proximal anastomosis in acute type A aortic dissection (ATAAD) surgery.

**Methods**. Patients who underwent ATAAD repair in our hospital between December 2020 and February 2023 were selected for inclusion in the study. The number of patients treated with the IHMS technique for aortic root repair were matched with those who received the conventional sandwich technique by the same surgeon. Perioperative and postoperative outcomes were analyzed and compared between the two groups.

**Results**. This study compared 44 patients in each group. The IHMS group had a shorter operation time than the sandwich group (6.07 h *vs.* 7.02 h, $p = 0.018$). The proximal anastomosis time (35.50 min *vs.* 40.00 min, $p = 0.013$), and extracorporeal circulation assistance time (70.00 min *vs.* 92.00 min, $p < 0.001$) were significantly reduced compared to the sandwich group. IHMS patients sustained less intraoperative blood loss (900 .00 mL *vs.* 1,500.00 mL, $p = 0.005$) and blood transfusion need (0 U *vs.* 0.75 U, $p = 0.028$) than patients in the sandwich group. Multivariate analysis revealed the IHMS technique to be independently associated with shorter suture time, less blood loss, and higher spontaneous heartbeat recovery. The IHMS group also had shorter durations of mechanical ventilation use, delirium, and hospital stay than the sandwich technique group. No statistically significant differences were found in postoperative morbidities during the follow-up period.

**Conclusion**. The IHMS technique for the aortic root anastomosis is simple, feasible and effective, particularly in ATAAD surgery with intimal rupture near the sinus-tubular junction to preserve the aortic valve during anastomosis of the ascending aorta.

## INTRODUCTION

Acute type A aortic dissection (ATAAD) is a life-threatening condition with a substantial risk of postoperative mortality (*Fang et al., 2022*). It is commonly heralded by a tear in

the ascending aorta above the sinus-tubular junction (STJ), resulting in the extension of the dissection flap into the sinus segment potentially leading to aortic regurgitation (*Movsowitz et al., 2000*; *Rylski et al., 2014*). Currently, proximal anastomosis remains technically challenging, and there is ongoing controversy regarding the most appropriate surgical strategy when the dissection involves the aortic root sinus (*Lai et al., 2003*). Given the fragility of the dissected aortic stump, conservative repairs may pose the risk of anastomotic disruption and life-threatening hemorrhage. Occasionally, proximal anastomosis is reinforced using a double-patch sandwich technique with prosthetic material (*Fleck et al., 2003*). However, this technique can prolong the aortic cross-clamp time or cause supravalvular obstruction. Other groups have proposed a direct reapproximating of the dissected layers and unsupported anastomosis at the STJ (*Yang et al., 2018*). However, when the dissection tear is close to the STJ, such conservative root preservation options can risk either bleeding or coronary blockage by prosthetic material. Here, we describe a modified technique for proximal anastomosis using an intermittent horizontal mattress suture (IHMS). This study aimed to evaluate the effects of the IHMS technique by assessing perioperative and postoperative outcomes and comparing them with those of the conventional sandwich method.

## MATERIAL AND METHODS

### Patients

This retrospective matched-controlled study was conducted in compliance with the Declaration of Helsinki. The study protocol was approved by the Medical Ethics Committee of Tongji Hospital on August 15, 2023 (Reference No.TJ-IRB20230806), which waived the requirement for informed consent owing to the anonymized nature of the data. Between December 2020 and February 2023, 303 consecutive patients underwent ATAAD repair. Among these patients, those who underwent IHMS aortic root repair surgery were independently identified from the in-hospital database by two cardiovascular physicians. All operations were performed by the same surgeon (Min Hu). Patients who underwent sandwich aortic root repair surgery were matched by age, sex, and body mass index (BMI) to those who underwent IHMS. The choice of the IHMS or sandwich technique was made by the medical team, individually judged on the basis of specific clinical presentation. The IHMS technique was selected when the aortic root intimal tear was low and close to the STJ.

The inclusion criteria were as follows: (1) ATAAD diagnosed by computed tomography angiography (CTA), with aortic root involvement; (2) no intimal tear in the aortic sinus; (3) aortic root diameter < 4 cm, or < 4.5 cm if > 65 years old; (4) no aortic annulus dilation and no obvious organic lesions in the aortic valvular lobe; (5) total arch replacement (TAR) and the frozen elephant trunk (FET) performed within 12 h of admission; (6) femoral artery used for arterial cannulation; and (7) unilateral antegrade cerebral perfusion. The exclusion criteria were as follows: (1) congenital connective tissue disease, such as Marfan syndrome, and (2) performance of additional procedures, such as aortic root replacement, coronary artery bypass grafting, valvuloplasty, or valvular replacement.

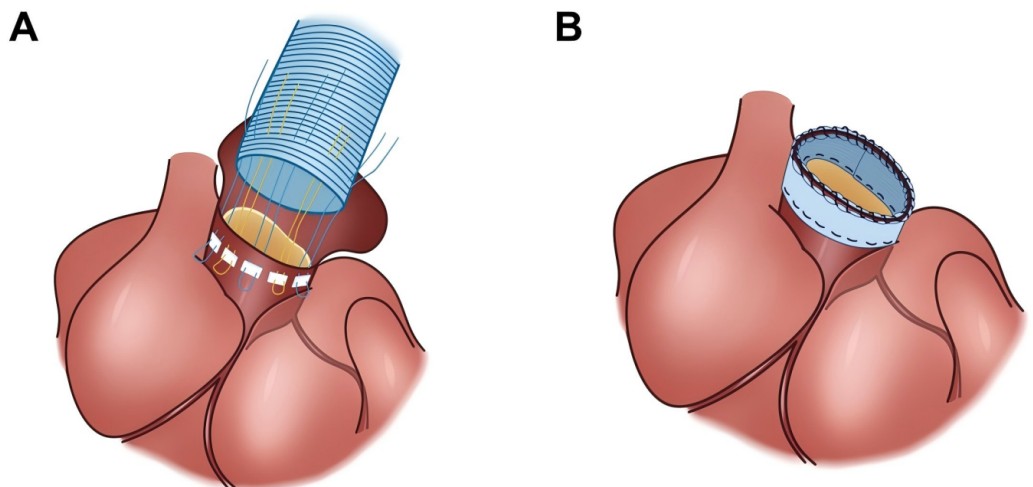

**Figure 1** **The comparison of proximal anastomosis of the aorta using the IHMS and double-patch sandwich techniques.** (A) The intermittent horizontal mattress suture for proximal anastomosis. (B) The sandwich technique for proximal anastomosis.

## Operative technique

The detail process of cardiopulmonary bypass, TAR, FET, and the double-patch sandwich technique for proximal anastomosis are described in Data S1. The IHMS technique for proximal anastomosis and bleeding management after repair are thoroughly described in the Supplemental Video S1. The comparison of proximal anastomosis of the aorta using the IHMS and sandwich techniques is shown in Fig. 1.

## IHMS technique

The IHMS technique for proximal anastomosis was performed as previously described (*Hu et al., 2024*) (Fig. 2). The aortic root was meticulously dissected above the coronary ostia. Subsequent removal of the thrombus from the false lumen was followed by horizontal trimming of the aortic wall at a distance of 5–8 mm above the STJ. Notably, the 1/3 posterior wall of the aorta remained continuous with that of the proximal aorta. This strategy aims to allow potential wrapping of this 1/3 posterior aortic wall tissue around the aortic root in cases where bleeding or oozing from the root is found after repair. The aortic valve was resuspended at the three commissures using a pledget 4–0 Prolene suture. Additionally, reinforcement of the aortic sinus wall with a pledget 5–0 Prolene suture was conducted in cases where dissection involved the coronary ostia.

At the end of circulatory arrest, we proceeded with the proximal anastomosis; the needle was entered through the adventitia of the aorta at the STJ and exited from the intima with a total of 5–6 stitches in each sinus using 2–0 polyester pledgeted suture. The sutures were secured with a suture anchoring coil device (Fig. S1). Subsequently, the Dacron graft was sutured with an inner-to-outer needle, followed by sheathing into the aortic root to tighten the suture and tie the knots. Equivalent and small stitch intervals were required to

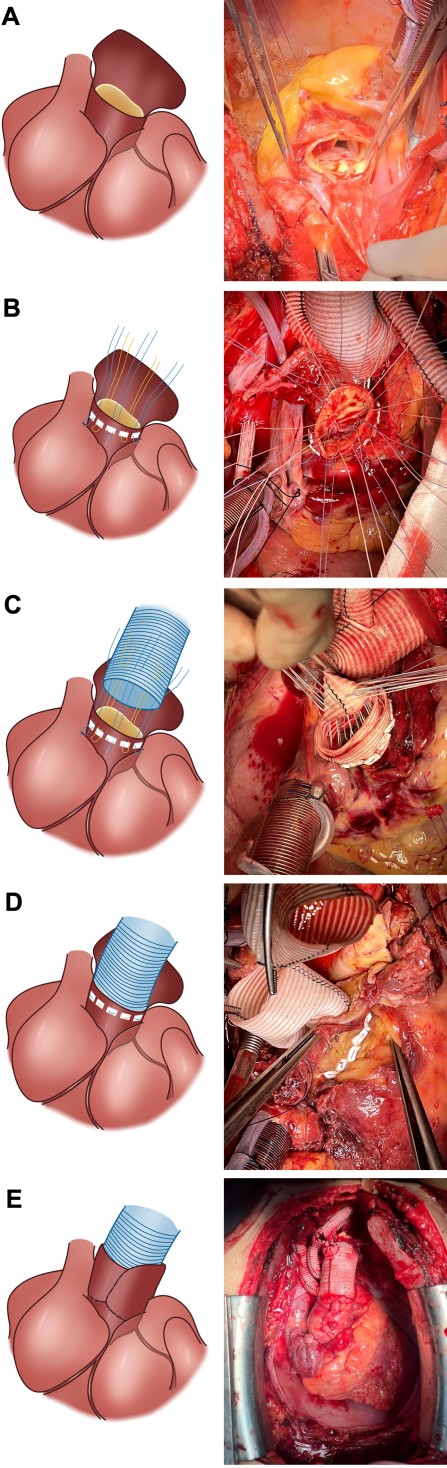

**Figure 2   Intermittent horizontal mattress suture technique in aortic root anastomosis.** (A) The 1/3 posterior wall of the aorta remains continuous with that of the proximal aorta. (B) The needle is inserted into the adventitia of the aorta at the sinus-tubular junction and released from the intima. (C) Inner-to-outer suturing of the Dacron graft; (D) Sheathing of the Dacron graft into the aortic root to tighten the suture and tie the knots. (E) The residual active wall is wrapped around the bleeding site.

minimize anastomotic bleeding. In patients with a frail aortic wall, pericardial or artificial vessel strips were added both inside and outside the aorta (Fig. S2).

Anastomosis of the proximal end of the four-branched artificial graft to the aortic root was achieved using 4–0 polypropylene. Subsequently, the aortic cross-clamp was released, enabling reconstruction of the left common carotid artery. After contractility recovered, the anastomosis was checked. If bleeding was present, additional sutures were applied. If this approach did not control the bleeding, the posterior aortic wall remnant was wrapped around the anastomotic bleeding site, or a fistula was created by wrapping the aortic root to the right atrial appendage using an eight mm Dacron tube.

## Definitions

The quantity of intraoperative bleeding was determined from the cardiopulmonary bypass suction volume. The time from the end of circulatory arrest to the end of the aortic cross-clamp was defined as the time of proximal anastomosis. Postoperative blood loss was assessed by monitoring suction drainage in the first 24 h. Indications for re-exploration due to bleeding were: (1) high drain output (>200 mL/h) during the initial 3 h post-operation; and (2) postoperative echocardiography revealing signs of pericardial tamponade. The circulation assistance time denoted the duration from the initiation of cardiac contraction until the discontinuation of all extracorporeal circulation systems.

## Postoperative treatment and follow-up

Postoperatively, all patients were transferred to the cardiac surgery intensive care unit. After weaning from the oxygen ventilator, patients were transferred to the general ward once they met the following criteria: oxygen saturation >96%, pleural drainage volume ≤100 mL/h, blood pressure maintained between 90/60 mmHg to 120/80 mmHg, heart rate maintained at 60–100 beats per minute (bpm), and appropriate neurological condition. All patients were advised to undergo CTA before discharge, at 3 months, 6 months, and 1 year after surgery, and annually thereafter.

Follow-up data were collected through telephone interviews or visits to the clinic. Mortality and aortic root reoperation data were collected as follow-up data.

## Outcomes

The primary outcome was the aortic root suture time. Other outcomes included aortic cross-clamp time, cardiopulmonary bypass time, spontaneous heartbeat recovery rate, operation time, intraoperative bleeding, postoperative complications, cardiac surgery intensive care unit time, extracorporeal circulation assistance time, and the mortality rate.

## Data collection and statistical analyses

SPSS software version 21.0 (IBM Corp., Armonk, NY, USA) was used for statistical analysis. Continuous variables with a normal distribution were compared using $t$-tests and data were expressed as mean ± standard deviation. Continuous variables deviating from a normal distribution were analyzed using Mann–Whitney $U$ tests and reported as median (interquartile range). Categorical variables were assessed using chi-squared and Fisher's exact tests and expressed as percentages. Linear regression analysis was performed to

investigate the associations between the suture method and aortic root suture time and intraoperative blood loss. Logistic regression analysis was performed to investigate the association between the suture method and spontaneous heartbeat recovery. A $p <0.05$ was considered statistically significant.

## RESULTS

### Preoperative characteristics

Between December 2020 and February 2023, 44 patients underwent the IHMS technique and were matched with 44 patients who underwent the sandwich technique (Fig. S3). The preoperative characteristics are shown in Table 1. There was no shock (blood pressure < 90/60 mmHg) before surgery in either group, and there were no significant differences in baseline characteristics between the two groups.

### Intraoperative outcomes

The intraoperative outcomes are shown in Table 2. The mean proximal anastomosis time in the IHMS group was shorter than the average time in the sandwich group (35.50 (inter quartile range: 30.00–43.00) min *vs.* 40.00 [35.00–52.00] min, $p = 0.013$), along with a reduced operation time (6.07 [5.31–7.19] h *vs.* 7.02 [6.01–7.57] h, $p = 0.018$) and extracorporeal circulation assistance time (70.00 [60.00–89.00] min *vs.* 92.00 [72.00–110.00] min, $p < 0.001$). The spontaneous heartbeat recovery rate was significantly higher in the IHMS group compared to the sandwich group (59% *vs.* 21%, $p < 0.001$). The mean intraoperative blood loss was 900.00 (600.00–1,500.00) mL in the IHMS technique group and 1,500.00 (800.00–2,225.00) mL in the sandwich technique group ($p = 0.01$). The average volume of intraoperative red blood cell transfusion in the IHMS group was also lower compared to the sandwich technique group (0 [0–2.00] U *vs.* 0.75 [0–4.88] U, $p = 0.028$). No patients had uncontrolled bleeding at the anastomotic site that required aortic root replacement.

Multivariate linear regression analysis revealed that the IHMS technique was independently associated with shorter aortic root suture time and lower intraoperative blood loss, while multivariate logistic regression analysis identified that the IHMS technique was independently associated with a lower risk of requiring electrical defibrillation for intraoperative heartbeat recovery (Table 3).

### Postoperative outcomes

Postoperative outcomes are shown in Table 4. The IHMS group had a shorter mean duration of mechanical ventilation (44.00 [18.00–79.25] h *vs.* 71.00 [35.25–144.50] h, $p = 0.008$), delirium (3 [1–6] days *vs.* 4.5 [2–9] days, $p = 0.040$), and hospital stay length (19 [16–24] days *vs.* 24 [15.25–35] days, $p = 0.015$) compared to those in the sandwich group. There were no significant differences in postoperative blood loss or the rate of re-exploration for bleeding (IHMS: 420.50 ± 353.79 mL *vs.* sandwich: 411.40 ± 291.86 mL, $p = 0.90$ and IHMS: 2% *vs.* sandwich: 2%, $p > 0.99$, respectively). There were no significant differences between the two groups in the rates of postoperative complications.

**Table 1  Preoperative characteristics of enrolled patients.**

| Categories | IHMS technique (n = 44) | Sandwich technique (n = 44) | p value |
|---|---|---|---|
| Age—years | 50.00 ± 10.34 | 49.00 ± 10.47 | 0.73 |
| Male | 31 (70.5) | 33 (75.0) | 0.63 |
| Body mass index—kg/m$^2$ | 25.60 ± 3.61 | 25.60 ± 4.14 | 0.97 |
| Diabetes mellitus | 1 (2.3) | 1 (2.3) | >0.99 |
| Hypertension | 37 (84.1) | 37 (84.1) | >0.99 |
| Cerebral disease history | 4 (9.1) | 3 (6.8) | 0.55 |
| Normal white blood cell | 16 (36.4) | 11 (25.0) | 0.25 |
| Anemia | 17 (38.6) | 14 (31.8) | 0.50 |
| Platelet $< 100 \times 10^9$/L | 3.00 (6.80) | 6.00 (13.60) | 0.48 |
| Elevated INR | 5 (11.40) | 2 (4.50) | 0.43 |
| Prothrombin time—s | 13.80 (13.30–14.90) | 13.60 (14.10–15.00) | 0.56 |
| Hypoproteinaemia | 10 (22.7) | 6 (13.6) | 0.27 |
| Elevated creatinine | 9 (20.5) | 9 (20.5) | >0.99 |
| Elevated triglyceride | 7 (15.9) | 9 (20.4) | 0.58 |
| Elevated total cholesterol | 2 (4.5) | 2 (4.5) | >0.99 |
| Transient ischemic attack | 6 (13.6) | 7 (15.9) | 0.76 |
| Left ventricle ejection fractions < 50% | 4 (9.1) | 2 (4.5) | 0.67 |
| Aortic valve regurgitation | 8 (18.2) | 7 (15.9) | 0.78 |
| Pericardial tamponade | 4 (9.1) | 2 (4.5) | 0.67 |
| Vessels involved by the dissection[*] | | | |
| Brachiocephalic trunk | 9 (20.5) | 5 (11.4) | 0.24 |
| Left common carotid artery | 3 (6.8) | 2 (4.5) | >0.99 |
| Left subclavian artery | 3 (6.8) | 2 (4.5) | >0.99 |
| Celiac trunk artery | 8 (18.2) | 10 (22.7) | 0.60 |
| Mesenteric artery | 10 (22.7) | 9 (20.5) | >0.99 |
| Renal artery | 15 (34.1) | 13 (29.5) | 0.65 |
| Lower extremity arteries | 3 (6.8) | 2 (4.5) | >0.99 |
| Shock | 0 (0) | 0 (0) | >0.99 |

**Notes.**
Data are presented as n (%), mean ± standard deviation, or median (interquartile range).
IHMS, intermittent horizontal mattress suture; INR, International Normalized Ratio.
*Vessels involved by the dissection which were confirmed by computerized tomographic angiography.

## Follow-up

The median follow-up time of the IHMS group was shorter than that of the sandwich group (21.00 [17.00–23.00] months *vs.* 23.50 [19.30–24.00] months, $p = 0.030$). Three patients in the IHMS group died at one, nine, and eleven months after surgery. Three patients in the sandwich group died, at one, five, and ten months after surgery. No patients required aortic root reoperation due to aortic root events during follow-up.

## The IHMS technique in patients with an intimal rupture site near STJ

It is noteworthy that in the IHMS group, eight patients had an intimal rupture site within a five mm range superior to the coronary artery origin (Fig. S4). All eight patients retained their original aortic valves and did not require re-exploration for bleeding. During more

**Table 2 Intraoperative characteristics of enrolled patients.**

| | IHMS technique (n = 44) | Sandwich technique (n = 44) | p value |
|---|---|---|---|
| PA time*—min | 35.50 (30.00–43.00) | 40.00 (35.00–52.00) | 0.013 |
| Operation time—h | 6.07 (5.31–7.19) | 7.02 (6.01–7.57) | 0.018 |
| CPB time—min | 204.50 ± 44.22 | 217.80 ± 35.32 | 0.071 |
| ACC time—min | 116.30 ± 19.32 | 116.50 ± 21.70 | 0.68 |
| CA time—min | 35.80 ± 5.82 | 33.40 ± 7.74 | 0.058 |
| Circulation assistance time—min | 70.00 (60.00–89.00) | 92.00 (72.00–110.00) | 0.007 |
| Red blood cell transfusion—U | 0 (0–2.00) | 0.75 (0–4.88) | 0.028 |
| Autologous transfusion—mL | 500.00 (250.00–750.00) | 500.00 (212.50–750.00) | 0.40 |
| Intraoperative urine volume—mL | 1,000.00 (462.50–1,500.00) | 1,250.00 (800.00–1,600.00) | 0.12 |
| Intraoperative blood loss—mL | 900.00 (600.00–1,500.00) | 1,500.00 (800.00–2,225.00) | 0.005 |
| Spontaneous heartbeat recovery | 26 (59.1) | 9 (20.5) | <0.001 |

Notes.
Data are presented as n (%), mean ± standard deviation, or median (interquartile range).
PA, proximal anastomosis; IHMS, intermittent horizontal mattress suture; CPB, cardiopulmonary bypass; ACC, aortic cross-clamp; CA, circulatory arrest.
*Proximal anastomosis time was defined as the time interval from the beginnings of restoration of cardiopulmonary bypass to the end of aortic cross-clam.

**Table 3 Multivariate regression analysis of association between intermittent horizontal mattress suture technique and intraoperative outcomes.**

| | β (95% CI)* | OR (95% CI)* | p value |
|---|---|---|---|
| Proximal anastomosis time—min | −6.493 [−11.642–1.524] | – | 0.011 |
| Intraoperative blood loss—L | −1.530 [−2.297–0.768] | – | <0.001 |
| Requiring electrical defibrillation for intraoperative heartbeat recovery | – | 0.161 [0.059–0.440] | <0.001 |

Notes.
OR, odds ratio; CI, confidence interval.
*Reference: conventional sandwich technique.

than 1-year of follow-up, no aortic events, such as aortic root dilation, stenosis, or aortic valve regurgitation, were observed in these eight patients (perioperative and follow-up data are presented in Tables S1–S2).

## DISCUSSION

This study aimed to investigate the efficacy of the IHMS technique for proximal anastomosis in ATAAD. The results suggest that the IHMS technique may offer advantages over the sandwich technique in terms of operative time, duration of proximal anastomosis, extent of cardiopulmonary bypass, intraoperative bleeding and blood transfusion, spontaneous heartbeat recovery, and durations of ventilator use, delirium, and hospital stay. The two-year follow-up results showed no pseudoaneurysm formation in the aortic root in the IHMS group. Aortic root events, reoperation rates, and mortality were also comparable between the two groups.

The management of ATAAD poses significant technical challenges owing to the fragility of the dissected aortic wall (*Fang et al., 2022*). Proximal anastomosis is a key component

**Table 4  Postoperative characteristics of enrolled patients.**

|  | IHMS technique (n = 44) | Sandwich technique (n = 44) | p value |
|---|---|---|---|
| Hospital stay time—d | 19.00 (16.00–24.00) | 24.00 (15.25–35.00) | 0.015 |
| Ventilation—h | 44.00 (18.00–79.25) | 71.00 (35.25–144.50) | 0.008 |
| Delirium time—d | 3.00 (1.00–6.00) | 4.50 (2.00–9.00) | 0.040 |
| First day drainage—mL | 342.50 (216.25–518.75) | 362.50 (150.00–593.75) | 0.90 |
| Length of ICU stay—d | 6.00 (5.00–11.00) | 8.00 (5.25–11.75) | 0.23 |
| Paraplegia | 0 (0) | 1 (2.3) | 0.32 |
| Renal failure | 2 (4.5) | 3 (6.8) | >0.99 |
| Stroke | 3 (6.8) | 4 (9.1) | >0.99 |
| Liver failure | 0 (0) | 1 (2.3) | 0.32 |
| Pulmonary infection | 15 (34.1) | 17 (38.6) | 0.66 |
| Septicemia | 1 (2.3) | 1 (2.3) | >0.99 |
| Gastrointestinal bleeding | 1 (2.3) | 2 (4.5) | >0.99 |
| Advanced life support | 1 (2.3) | 0 (0) | >0.99 |
| Re-exploration for bleeding | 1 (2.3) | 1 (2.3) | >0.99 |

**Notes.**
Data are presented as n (%), mean ± standard deviation, or median (interquartile range).
IHMS, intermittent horizontal mattress suture; ICU, intensive care unit; IQR, interquartile range.

of ATAAD surgery (*Shrestha, Haverich & Martens, 2017*) and several techniques have been developed to avoid bleeding, pseudoaneurysm formation, or aortic regurgitation. The use of Teflon strips and biological glue to reinforce proximal aortic anastomoses is well established; however, the use of Teflon felt in aortic surgery has been associated with extensive inflammation and adhesions, potentially impeding complete healing of the dissected aorta (*Yang et al., 2018*). The application of biological glue to achieve hemostasis may increase the risk of aortic root re-dissection and the occurrence of embolic events in the coronary and cerebral arteries (*Haijima et al., 2022*). The traditional double-patch sandwich technique involving Teflon felt is a widely used in proximal anastomosis (*Fleck et al., 2003*). Nevertheless, substantial rates of late re-intervention have been reported after performing the sandwich technique with Teflon felt (23%) and biological glue (20%), respectively (*Concistrè et al., 2012*; *Yamanaka et al., 2012*; *Gocołet al., 2021*; *Haijima et al., 2022*).

Considering the complexity of the conventional sandwich technique, we adopted the IHMS technique using thin 2–0 polyester pledgeted sutures without Teflon felt or biological glue in September 2021. The telescopic anastomosis technique (*Lin et al., 2023*), akin to the IHMS technique, does not require the use of Teflon felt and exhibits stabilizing effects, preventing dilation of the aortic root and mitigating the progression of aortic valve regurgitation. However, the telescopic anastomosis technique places pledgets on the inner wall of the aorta to prevent sutures cutting the aorta, which may predispose to thrombosis and infection. The rapid blood flow at the proximal ascending aorta impacting against the rough surface formed by the pledgets can lead to the destruction of red blood cells, potentially causing hemolysis (*Sakaguchi & Takano, 2016*). In contrast, the IHMS technique involves everting the proximal end of the prosthetic vessel and suturing it within

the patient's native aortic root. The everted prosthetic vessel acts as a cuff, enabling more uniform and precise tension distribution during knot tying, and eliminating the risk of cutting injuries to the fragile vascular wall.

The IHMS technique was developed to mitigate bleeding within the ascending aorta after anastomosis due to the felt stiffness and incomplete hermetic suture, which we frequently observed in the conventional sandwich approach. Due to the large felt coverage area, when blood spread through the felt, the bleeding site was difficult to distinguish, increasing the difficulty of hemostasis, and occasionally even resulting in extensive root bleeding. In the IHMS technique, the use of a 2–0 polyester pledgeted suture evenly distributes the tension and eliminates the cutting effect of the suture on the fragile aortic wall. The lack of Teflon felt allowed visualization of the bleeding site and satisfactory hemostasis in the IHMS approach. Bleeding from the needle hole can be stopped by using the additional sutures or the remnant of the aortic wall to cover the anastomosis. Patients in the IHMS group showed significantly less intraoperative blood loss compared to patients in the sandwich group. Additionally, the IHMS technique was independently associated with lower blood loss volumes.

If the aortic root intimal tear is low and close to the STJ, the coronary opening may be blocked when using the sandwich technique and cause serious consequences. The Bentall or David procedures are often the only choice in this situation. The former has anticoagulant-related risks, whereas the latter is complex and time-consuming, making it unsuitable for emergency surgery, especially when surgical experience is lacking. Using the IHMS technique, even if the suture site is close to or in the STJ, the opening of the coronary artery will not be obscured, ensuring myocardial blood supply. In addition, the IHMS technique is extremely similar to suture placement in conventional valve replacement and can thus be performed by junior surgeons trained in valve replacement surgery. The generalizability of this approach during the procedure allows for reproducibility in the surgical field. The intimal tear was located at the STJ, approximately five mm above the coronary artery, in eight patients in the IHMS group , for whom satisfactory results were achieved. The rate of spontaneous heartbeat recovery in the IHMS group was significantly higher than that in the sandwich group and was independently associated with the IHMS technique. These results indicate that coronary blood flow was not obstructed, and the myocardial blood supply was sufficient after the ascending aorta was opened.

A long operation time is associated with postoperative complications and mortality in patients with ATAAD (*Conzelmann et al., 2016*). In our study, compared with the sandwich approach group, the IHMS technique group exhibited shorter proximal anastomosis time. Furthermore, a higher proportion of patients in the IHMS technique group experienced spontaneous intraoperative heartbeat recovery, and there was better hemostasis at the aortic root anastomosis site, thereby reducing the need for additional sutures and saving time. Collectively, these factors contributed to shorter overall surgical and anesthesia durations in the IHMS technique group compared with the control group. Short operative and anesthesia times have been associated with improved postoperative mental recovery in patients (*Shim et al., 2019*). A recent study found that a longer duration of surgery is a significant risk factor for postoperative delirium in patients with ATAAD

(*Lu et al., 2024*). An increase in surgical duration corresponds to an increased duration of anesthesia. Prolonged exposure to anesthetic agents such as fentanyl and propofol, that act on cholinergic neuronal receptors, can lead to inhibition of cholinergic neural effects and an exacerbation of postoperative delirium in patients (*Ahn & Bang, 2022*).

When there is persistent bleeding at the suture site of the aortic root, the IHMS approach utilizes the remaining residual aortic wall to create a wrap, a procedure similar to the classical Cabrol shunt (*Raghuram et al., 2023*), but with a key difference: the connection to the right atrium is only established when the bleeding at the root cannot be controlled by other methods. Furthermore, unlike the "Mantle-style" modification of the Cabrol shunt (*Lin, Tsai & Wu, 2019*), in IHMS technique procedure, the posterior wall of the aorta is not anastomosed with a graft or other cardiac tissue structures. Instead, hemostasis is achieved through the compressive force exerted by the tight wrap of the aortic posterior wall and the coagulation of blood that occurs at the wrap site following the leakage.

In conventional sandwich root anastomosis, felt pieces are present at the anastomosis site, which may increase the risk of adhesions and inflammation (*Yang et al., 2018*). Moreover, the reduced diameter of the aortic root may be associated with aortic stenosis (*Matsuura et al., 2004*). In addition, in the IHMS technique, the anastomosis can reach the STJ and the false lumen within the ascending aorta is completely eliminated. This maintains the natural geometry of the aortic root, supporting aortic valve function and facilitating the insertion of the aortic sinus intima and adventitia, aiding closure of the aortic dissection. Immediate postoperative transesophageal echocardiography and CTA confirmed that the repaired false lumen of the aortic sinus completely disappeared in all IHMS group patients. The aortic sinus diameter in the IHMS technique group did not significantly increase during follow up, and most preoperative aortic regurgitation was diminished. Although this study has achieved certain advances in performing aortic root anastomosis using the IHMS technique, given the complexity and variability of aortic dissection cases that may influence surgical outcomes, we encourage future researchers to conduct larger-scale studies using this method to further validate the advantages of the IHMS technique in managing the aortic root.

This study has certain limitations, including a small sample size, restricted follow-up period, single-center experience and absence of randomization. These factors, along with an observational study design, may introduce patient and technique selection bias, restricting the generalizability of the conclusions. The location of intimal tear was not included as a matching criterion. The position of the intimal tear reflects the severity of aortic dissection, which can affect the patient's postoperative prognosis. However, no significant difference of the preoperative location of aortic intimal tear was observed between the two groups.

The preparation time for trimming, dissecting, and suspending the aortic root was not included in the proximal anastomosis time, which may result in the underestimation of relevant operation time. Furthermore, pseudoaneurysms at the anastomosis typically develop within an average of 40 months (ranging from 1–65 months) after surgery (*Suzuki et al., 2006*). Therefore, a longer follow-up period is required to investigate the occurrence of pseudoaneurysms after IHMS.

## CONCLUSIONS

The IHMS technique for the aortic root anastomosis in ATAAD is simple, feasible, and effective, and is especially suitable for beginners or centers without adequate surgical experience. This technique has shown promising early outcomes, even with the presence of intimal rupture near the STJ, and can preserve the native aortic valve, ultimately bringing greater benefits to patients with aortic dissection.

## ACKNOWLEDGEMENTS

We thank Professor Eduard Quintana (from the University of Barcelona in Spain) for polishing our article.

### Funding

Min Hu was supported by the Hubei Provincial Natural Science Foundation (grant number 2023 AFB672). The funders had no role in study design, data collection and analysis, decision to publish, or preparation of the manuscript.

### Grant Disclosures

The following grant information was disclosed by the authors:
Hubei Provincial Natural Science Foundation: 2023 AFB672.

### Competing Interests

The authors declare there are no competing interests.

### Author Contributions

- Jiajun Li conceived and designed the experiments, performed the experiments, analyzed the data, prepared figures and/or tables, authored or reviewed drafts of the article, and approved the final draft.
- Yongzhi Zhou performed the experiments, prepared figures and/or tables, authored or reviewed drafts of the article, and approved the final draft.
- Yucong Zhang performed the experiments, analyzed the data, prepared figures and/or tables, and approved the final draft.
- Xuegui Chen performed the experiments, authored or reviewed drafts of the article, and approved the final draft.
- Jing Wang performed the experiments, authored or reviewed drafts of the article, and approved the final draft.
- Xiang Wei conceived and designed the experiments, authored or reviewed drafts of the article, and approved the final draft.
- Min Hu conceived and designed the experiments, authored or reviewed drafts of the article, and approved the final draft.

## Human Ethics

The following information was supplied relating to ethical approvals (i.e., approving body and any reference numbers):

The study protocol was approved by the Medical Ethics Committee of Tongji Hospital on August 15, 2023 (Reference No. TJ-IRB20230806).

## Data Availability

The raw data are available in the Supplemental File.

## Supplemental Information

Supplemental information for this article can be found online at http://dx.doi.org/10.7717/peerj.19159#supplemental-information.

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
