# Peer review of "Intermittent horizontal mattress suture in proximal anastomosis for acute type A aortic dissection: a retrospective study"

_PeerJ, doi:10.7717/peerj.19159_

## Round 0.1 · original submission · Major Revisions

Please response to the reviewers point by point.

·

Basic reporting

The authors use appropriate technical terminology, and the manuscript maintains a formal and professional tone. The structure of sentences is mostly clear. Here are my comments:

- Some sentences are long and complex, making them difficult to follow. Specifically, this sentence needs revision: "Currently, proximal anastomosis remains technically challenging, in the setting of acutely dissected aortic root sinuses it remains debatable which operative strategy is the most appropriate."
- Ensure that the terminology used is consistent throughout the manuscript. For instance, "operation time" vs. "surgical time".
- Please check for consistent use of articles.
- Some phrases could be more concise. "significantly higher in the IHMS group than in the sandwich group" could be changed to "significantly higher in the IHMS group compared to the sandwich group."
Overall, thorough proofreading by a fluent English speaker or an editing service would help address the grammatical inconsistencies and improve the manuscript's readability.
- It would be a good idea to include visual comparisons between the two techniques (only the IHMS is included).

Experimental design

In the Methods section, the authors stated that patients were matched based on age, sex, and BMI. However, there is no mention of matching based on the location of the intimal tear, which can be a significant factor in aortic dissection surgeries. The authors should acknowledge this limitation in their Discussion section. They should address how differences in the location of the intimal tear might have influenced their results.

Validity of the findings

I encourage the authors to recommend future studies with more patients to validate their findings. This is particularly important given the complexity and variability of aortic dissection cases, which may affect surgical outcomes.

Some significant differences between the two groups, such as the reduced duration of delirium in the IHMS group, are mentioned in the Results section but not adequately discussed. The authors should provide a more thorough discussion of why these differences might exist, how they relate to the IHMS technique, and what clinical implications they have.

Reviewer 2 ·

Basic reporting

1. This article is generally well-written and easy to read. However, the authors should seek thorough professional English editing before publication, including for supplementary files.

2. Please ensure that all figures are oriented identically. Specifically, rotate the surgical pictures so that the upper part of each image indicates the patient's head.

3. In Table 2, the expression of data for the PA time and circulation assistance time differs between the IHMS and sandwich groups; one is presented as median (IQR) while the other is presented as mean ± SD. Similar discrepancies exist in the results section of the main text (lines 164, 166, 182), as well as in Table 4 (length of hospital stay and ICU stay). Please correct these inconsistencies.

4. In Table 3, I recommend using two separate columns to present the beta value and OR (95% CI).

Experimental design

no comment

Validity of the findings

1. A group from Taiwan has developed a similar anastomosis technique using multiple mattress sutures for acute type A aortic dissection. Please add a comparison of these two techniques and include the citation: Lin TW, Wu HY, Tsai MT, Hu YN, Wang YC, Roan JN, Luo CY, Kan CD. "Aortic root remodeling after surgical repair of acute type A aortic dissection using different anastomosis techniques." JTCVS Tech. 2023 Jul 22;21:18-25. doi: 10.1016/j.xjtc.2023.06.018.

2. The residual aortic wall wrapping method appears to be a modification of the classical Cabrol shunt. Please include a brief discussion of the differences. Additionally, did you secure the aortic wall to the graft distally with sutures? If so, this modification conceptually aligns with the work of the same group from Taiwan. Please reference: Lin TW, Tsai MT, Wu HY. "Mantle-style" modification of Cabrol shunt for hemostasis after extended aortic reconstruction in acute type A aortic dissection. Gen Thorac Cardiovasc Surg. 2019 Nov;67(11):1001-1005. doi: 10.1007/s11748-019-01151-1. If sutures were not used to secure the wrapping aortic wall to the graft, how was hemostasis achieved?

3. Please clarify the definition of “circulation assistance time” in the methods section, particularly as there was a statistical difference noted between the IHMS and conventional groups in this study.

4. The proximal anastomosis time was defined as the interval from the end of circulatory arrest to the end of the aortic cross-clamp. While this definition is reasonable, it may introduce bias. Personally, I adopt a "distal-first" principle in aortic surgery, performing the distal anastomosis once the target temperature for deep or moderate hypothermia is achieved, then re-warming and resuming full body flow before performing the proximal anastomosis. During the cooling period before the distal anastomosis, I also prepare the proximal aortic end (e.g., completing the sandwich with Teflon felts). Therefore, I suggest that the total time for the proximal procedure should encompass both the sandwich preparation and the subsequent anastomosis. Did you place mattress sutures on the proximal aorta before or after circulatory arrest? Please provide a more detailed description of your process in the operative technique section. If the parameter “proximal anastomosis time” does not include any preparation of the proximal aorta before circulatory arrest, I recommend acknowledging this potential bias in the limitations section.

5. The authors argue that the use of Teflon felt in the sandwich technique may lead to complications at the anastomosis site, such as stenosis or thrombus formation. Is this concern based on the authors’ experience, or is there literature to support it? The sandwich technique described by the authors involves placing Teflon felt both inside and outside the aortic wall, as mentioned in the supplementary document. While variations exist based on institutional preferences, a more common approach is to place either a Teflon felt or a synthetic patch into the false lumen, with felt positioned outside the adventitia. This method ensures that no artificial material remains in the aortic lumen post-anastomosis, potentially minimizing the risk of stenosis or thrombus formation at the site.

6. The authors mention using “vessel strips” to support the fragile aortic wall as needed. The operative images show that these strips are rings of Dacron graft. When placing pledged-mattress sutures at the proximal aorta, did you also pass these sutures through the vessel strips? Please describe this technique in greater detail in the operative technique section of the supplementary document. I believe that using vessel strips to strengthen the aorta is a modification of the sandwich technique, employing a Dacron graft ring instead of Teflon felt.

Additional comments

The authors should be commended for their innovative anastomosis technique used in operations for acute type A aortic dissection. Given that surgical treatment for this condition remains high-risk in the current era, every attempt to improve surgical outcomes is encouraged. This article provides readers with informative insights into this potentially lethal disease.

---

## Round 0.2 · Minor Revisions

Please respond to the reviewer's remaining issues point by point.

·

Basic reporting

I have no additional comments.

Experimental design

I have no additional comments.

Validity of the findings

I have no additional comments.

Reviewer 2 ·

Basic reporting

The authors have adequately addressed the reviewer’s questions and revised the manuscript.

Experimental design

no comment

Validity of the findings

My only remaining concern pertains to Table 3:

1. The column header “β/OR (95% CI)”—should this be written as “β (95% CI)” if the authors have separated β and OR into two columns?

2. In the table, the OR for spontaneous intraoperative heartbeat recovery (IHMS compared to the conventional sandwich technique) is shown as “0.161 (0.590 - 0.440)”. Could the authors clarify whether this OR indicates that IHMS results in a lower likelihood of spontaneous heartbeat recovery, or if it reflects a reduced likelihood of "failure" in spontaneous heartbeat recovery?

3. Furthermore, why does the confidence interval (0.59–0.44) in this table not encompass 0.161?

Additional comments

I believe this work is ready for publication once the aforementioned concerns are addressed.

---

## Round 0.3 · accepted · Accept

All the comments were fully addressed. Please make the final changes requested by Reviewer 1 during the proofing stage.

·

Basic reporting

Please remove the following part from the first sentence of the results in the abstract: 'as determined by [ median/average ] value comparison with Mann3Whitney U tests.' Also please remove 'In the same analysis,' from the second sentence.

Experimental design

no comment

Validity of the findings

no comment

Additional comments

no comment